# Finnish Retirement and Aging Study: a prospective cohort study

Sari Stenholm [1,2,3] Kristin Suorsa,[1,2] Tuija Leskinen [1,2] Saana Myllyntausta,[4] Anna Pulakka [5,6] Jaana Pentti,[1,2,7] Jussi Vahtera [1,2]

[1]Department of Public Health, University of Turku and Turku University Hospital, Turku, Finland
[2]Centre for Population Health Research, University of Turku and Turku University Hospital, Turku, Finland
[3]Research Services, Turku University Hospital and University of Turku, Turku, Finland
[4]Department of Psychology and Speech-Language Pathology, Turun Yliopisto, Turku, Finland
[5]Research Unit of Population Health, Faculty of Medicine, University of Oulu, Oulu, Finland
[6]Population Health Unit, Finnish Institute for Health and Welfare, Helsinki, Finland
[7]Clinicum, Faculty of Medicine, University of Helsinki, Helsinki, Finland

**Correspondence to**
Dr Sari Stenholm;
sari.stenholm@utu.fi

## ABSTRACT

**Purpose** The Finnish Retirement and Aging (FIREA) Study was set up to study changes in health behavioural and cardiometabolic risk factors across retirement transition, and to examine the long-term consequences of work and retirement on health and functioning with advancing age.

**Participants** Public sector workers whose estimated statutory retirement date was in 2014–2019 were invited to participate by sending them a questionnaire 18 months prior to their estimated retirement date. In the first phase of the FIREA Study, participants were followed up with annual surveys, accelerometer and clinical measurements during retirement transition into post-retirement years. The FIREA survey cohort includes 6783 participants, of which 908 belong also to the activity substudy and 290 to the clinical substudy.

**Findings to date** Collected data include survey measures about health, lifestyle factors, psychosocial distress, work-related factors as well as retirement intentions. Accelerometer and GPS devices are used to measure 24-hour movement behaviours. Clinical examination includes blood and hair sample, measurements of anthropometry, cardiovascular function, physical fitness, physical and cognitive function. Our results suggest that in general retirement transition seems to have beneficial influence on health behaviours as well as on physical and mental health, but there are large individual differences, and certain behaviours such as sedentariness tend to increase especially among those retiring from manual occupations.

**Future plans** The second phase of the FIREA Study will be conducted during 2023–2025, when participants are 70 years old. The FIREA Study welcomes research collaboration proposals that fall within the general aims of the project.

## INTRODUCTION

The Finnish Retirement and Aging (FIREA) Study was established in 2013 to understand in detail the short-term and long-term effects of work and retirement, an important life transition in late midlife, on people's health behaviours and health. Specifically, the FIREA Study was initiated (1) to examine changes in health behaviours, cardiometabolic risk factors and health indicators during retirement transition by following older workers from final working years to full-time retirement; (2) to study individual-related and work-related factors that predict working

---

## STRENGTHS AND LIMITATIONS OF THIS STUDY

⇒ The unique feature of the Finnish Retirement and Aging (FIREA) Study is that the data collection was targeted around estimated statutory retirement date and it continued annually from final working years into actual retirement.

⇒ In addition to survey measures, which are prone to response bias, participants have been followed with accelerometer and clinical measurements, which are more accurate and enable capturing subtle changes in movement behaviours and cardiometabolic risk markers.

⇒ The extensive Finnish national registers offer a possibility to complement collected data with information on participants' health, use of healthcare services as well as on neighbourhood characteristics.

⇒ The participation rate in the first phase of the FIREA Study has been good, especially among those who participated in the activity (≥88% across measurements years) and the clinical substudy (≥92% across measurements years).

⇒ The main limitation of the FIREA Study is a healthy worker effect as the inclusion criteria include participation in the working life after the age of 60 (those retired due to illness, disability or unemployed are not included in the study population); moreover, men and individuals living alone are under-represented, which further limits the generalisability of the results.

---

life participation in people aged 60 years and older and (3) to examine long-term consequences of work and retirement on health and functioning with an advancing age.

Transitioning from working life into retirement brings flexibility to daily time use, while work-related stressors are also removed, which may have extensive impact on health behaviours and social activities and consequently on health and well-being. At the time FIREA Study was initiated, several studies on retirement and health had been published,[1–4] and there were several large cohort studies following up participants from age 50 onwards, such as the Health and Retirement Study in the USA,[5] the English Longitudinal Study on Aging in the UK[6] and the Survey of

Health, Ageing and Retirement in Europe from several European countries.[7] However, only a few studies had studied changes in health behaviours by following individuals through retirement transition. These studies had reported changes in health behaviours, such as increased physical activity and improved sleep,[8–12] but the results were contrasting, and information on health and health behaviours was based on self-reported measures. To identify subtle changes in health behaviours and clinical risk markers, and to overcome measurement bias related to survey measures, the FIREA Study took one step further by establishing annual device-based and clinical measurements for older workers close to their estimated retirement data. These measurements have been complemented with annual surveys, including questions about lifestyle factors, physical and mental health, work and social factors, as well as with register-based information on participants' living environment and linkage to national health registries. The aim is to continue following up the study participants until old age.

## COHORT DESCRIPTION
### Study population and data collection
The eligible study population of the FIREA cohort comprised of public sector employees who were working in any town in Southwest Finland or in any of the a priori selected 10 towns or five hospital districts around Finland in 2012 and whose estimated statutory retirement date was between 2012 and 2025 (n=47 581). This register cohort was acquired from the pension insurance institute for the municipal sector in Finland (Keva), and included information about workers' date of birth, work history and estimated retirement date. Individuals whose estimated retirement date was between 2014 and 2019 were invited to participate in the FIREA Study by sending them a questionnaire 18 months prior to their estimated retirement date (n=10 629). The questionnaire included information sheet and informed consent. The data collection started in the fall 2013. In the first phase of the FIREA Study, questionnaires were sent once a year with an aim to gather data from at least two time points before and two time points after the transition to statutory retirement. In total, 6783 participants had responded to at least one questionnaire by the end of 2018. 5506 participants responded at least in two time points and of them 3948 provided response before and after retirement. Flow chart of the FIREA Study is shown in figure 1, and schematic overview of the data collection in figure 2.

In 2014, a FIREA activity substudy was launched and Finnish-speaking FIREA survey cohort participants who were still working and had an estimated statutory retirement date between 2016 and 2019 were invited to participate by mailing them a separate information sheet and informed consent (n=2663). Of them, 908 consented to participate and they were sent the accelerometer on average 2.1 months (SD 1.2) after completing the questionnaire. They were then followed via accelerometer measurements and survey questionnaires annually, at the same time of the year, up to five times in total.

One year later in 2015, a FIREA clinical substudy was initiated. Finnish-speaking survey cohort participants who were still working had an estimated statutory retirement date between 2017 and 2019 and who lived in the Southwest Finland were invited to participate by mailing them a separate information sheet and informed consent (n=773). Of them, 290 consented to participate

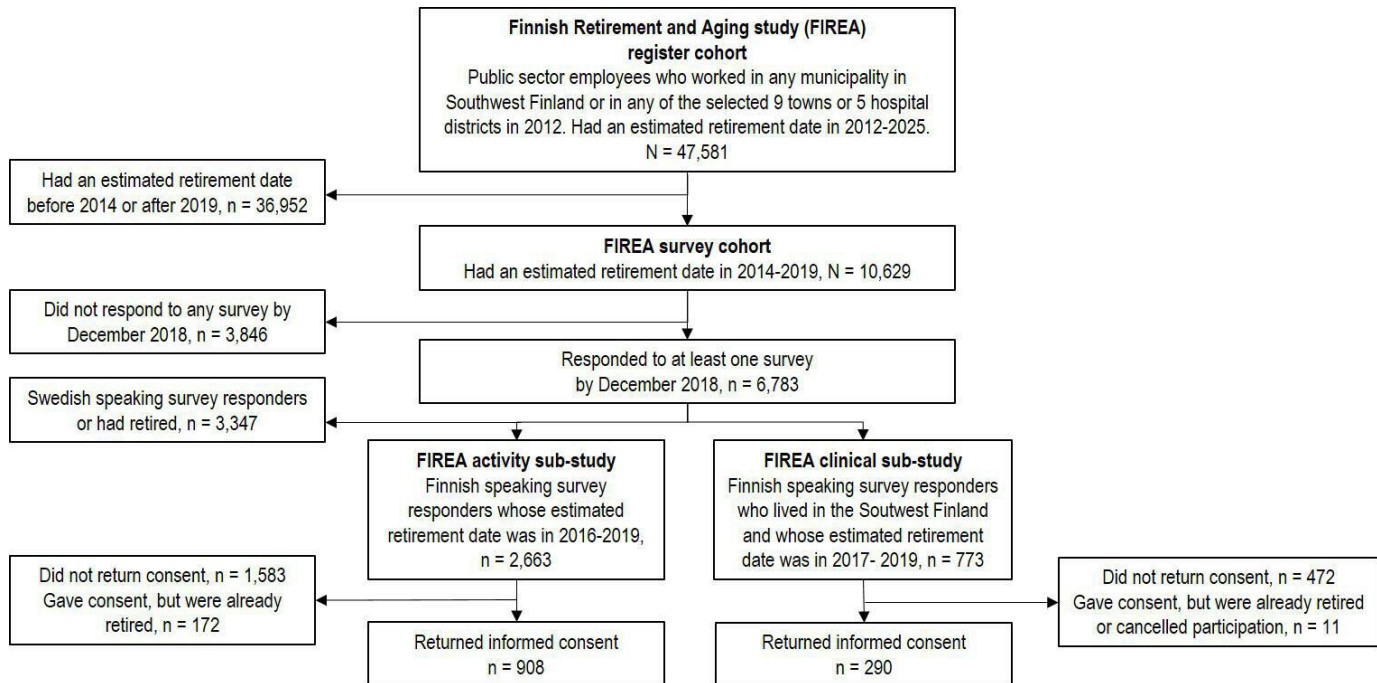

**Figure 1** Flow of the Finnish Retirement and Aging Study (FIREA).

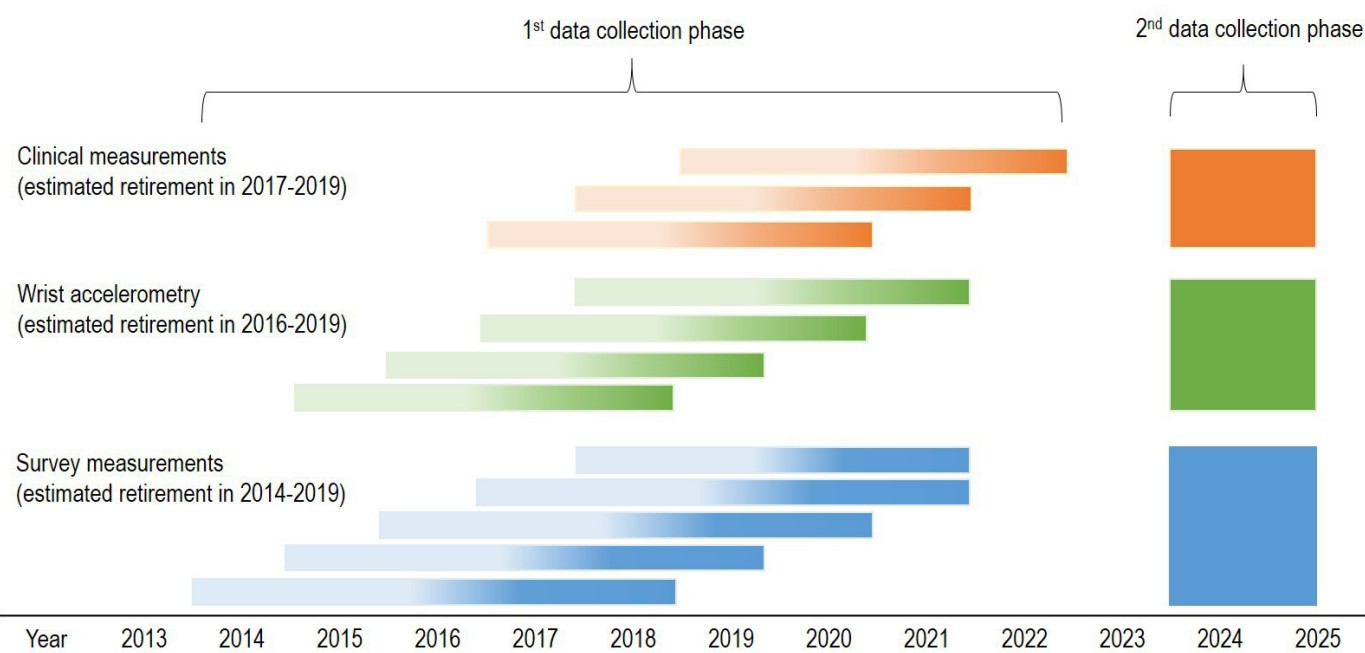

**Figure 2** Schematic overview of the data collection in the Finnish Retirement and Aging (FIREA) Study. Bar colour intensity indicates the working status of the participants. Light colour=full time working, dark colour=retired.

in annual clinical examination and the first visit took place on average 4.3 months (SD 3.4) after completing the questionnaire. Up to five annual follow-up visits were conducted until the participants had retired. Participants did not receive any reimbursement for taking part in the study.

Due to COVID-19 pandemic and consequent national restrictions, the data collection for activity and clinical substudy were temporarily discontinued between March and June in 2020. However, this influenced only a small proportion of the participants, since the majority of the participants had already retired and completed the first phase of the data collection before 2020.

Table 1 provides comparison between the participants in the survey cohort, the activity substudy, the clinical substudy and the non-participants. In comparison to the

**Table 1** Baseline characteristics of the Finnish Retirement and Aging (FIREA) Study participants and non-participants

| | Survey cohort (n=6783) | Activity substudy (n=908) | Clinical substudy (n=290) | Non-participants (n=3846) |
|---|---|---|---|---|
| Age (years), mean (SD) | 62.7 (1.2) | 62.4 (1.1) | 62.4 (1.0) | 62.7 (1.2) |
| Female, % | 82 | 85 | 83 | 75 |
| Occupational group, % | | | | |
| Manual | 40 | 33 | 31 | 45 |
| Lower non-manual | 30 | 29 | 34 | 28 |
| Upper non-manual | 30 | 38 | 35 | 26 |
| Married/cohabiting, % | 71 | 71 | 72 | |
| Physical activity (MET-hours per week),* mean (SD) | 23.6 (20.2) | 24.8 (20.0) | 28.8 (21.7) | |
| Body mass index (kg/m$^2$),* mean (SD) | 26.9 (4.6) | 26.6 (4.7) | 26.0 (4.0) | |
| Current smoking, % | 11 | 6 | 5 | |
| Alcohol risk use,† % | 3 | 2 | 3 | |
| Good or rather good self-rated health,‡ % | 73 | 77 | 82 | |

*Self-reported information.
†Alcohol risk use: >24 units/week for men and >16 units/week for women.
‡Self-rated health was assessed by asking participants to rate their overall health status on a 5-point scale (1=good, 2=rather good, 3=average, 4=rather poor, 5=poor).
MET, metabolic equivalent.

FIREA participants, the proportions of men and manual workers were greater among the non-participants. Participants in the survey cohort, the activity substudy and the clinical substudy did not differ from each other in terms of age, sex and marital status, but the proportions of manual workers and current smokers were greater in the survey cohort compared with activity substudy and clinical substudy participants. Self-reported physical activity was highest and self-rated health better among the clinical substudy participants and lowest and less optimal, respectively, in the survey cohort.

In the first phase of the FIREA Study (2013–2022), the data collection took place annually, always at the same time of the year. On average, the participants provided survey data on 3.6 measurement points (range 1–7, SD 1.4). The mean number of annual measurements in the activity substudy was 3.7 (range 1–5, SD 0.7) and in the clinical substudy 3.0 (range 1–5, SD 0.6).

Of the initial survey cohort, 58% have both pre-retirement and post-retirement data (n=3948), because many of the initial cohort had already retired when they were contacted for the first time. The actual retirement date was obtained from the survey and this information was used to identify pre-retirement and post-retirement measurements. Respective proportions for the pre-retirement and post-retirement measurements are 83% for the activity substudy and 88% for the clinical substudy. The higher proportions compared with the survey cohort reflect the fact that only those who were still working were invited to participate in these substudies.

### Retirement system in Finland

In Finland, from 2005 onwards, public sector employees have been able to retire on a statutory basis after the age of 63 years, but at the latest before the age of 68 years. Following a pension reform in 2017, each age group has their own retirement age, which is tied to the life expectancy. Postponing retirement from the estimated retirement age will accrue pension income level. After the pension reform, employees may also retire on partial early old-age pension, years of-service pension (for those who have worked for at least 38 years in job that requires great mental or physical effort), or on full or partial disability pension.[13] The FIREA Study participants were, due to the inclusion criteria, mostly retiring due to old age with a mean age of 64.0 years (SD 1.4).

### Measurements

#### Survey measurements

In the first phase of the FIREA study, annual surveys included questions about health, chronic diseases, physical functioning, memory, lifestyle factors, psychosocial distress, social networks, work-related physical and psychological stressors as well as retirement intentions and actual retirement date. The questionnaire was 20 pages long and it took about 20–30 min to complete. The content of the questionnaire was similar in each follow-up year. Details of the survey instruments are provided in online supplemental table 1. In addition to the multiple-choice questions, the survey included open-end questions regarding positive expectations and fears about retirement for those who were still working, and about positive and negative consequences of retirement for those who had retired.

#### Activity substudy measurements

Participants' 24-hour movement behaviours were measured using wrist-worn triaxial ActiGraph wActiSleep-BT and wGT3X-BT accelerometers (ActiGraph, Pensacola, Florida, USA), which were mailed to the participants. Participants were asked to wear the accelerometer on their non-dominant wrist for seven consecutive days and six consecutive nights at all times and fill in a diary of their workdays and days off, daily bedtimes and modes of commute. Details of the FIREA accelerometer measurement protocol have been described earlier.[14] Moreover, accelerometer data reduction procedures have been described in detail earlier.[14–17]

#### Clinical substudy measurements

Annual clinical examination visits were conducted at the University of Turku, in the Faculty of Medicine's research facilities. Measurements were conducted in the morning by trained study nurses and the visit lasted about 2.5–3 hours. Participants received feedback of the measurements for which results were available.

A 10-hour fasting blood sample were drawn by venipuncture in the morning shortly after arriving to the clinic. All blood samples were aliquoted at the same day and then stored at −80°C. Plasma cholesterol (total, low-density lipoprotein cholesterol, high-density lipoprotein cholesterol), triglycerides, fasting insulin and glucose and C reactive protein were analysed later at the laboratory of Turku University Hospital, Finland.

For the hair sample, a strand of hair was cut from a standardised area of the posterior vertex region of the head as close to the scalp as possible.[18] Hair samples were stored in foil in a dry place protected from light. Hair cortisol and cortisone were analysed later using mass spectrometry (Technical University of Dresden, Germany).

For anthropometrics, weight, height, body mass index and waist circumference were measured by using standard methods. Body composition was measured with Inbody 720 (Biospace, Seoul, Korea).[19] Systolic and diastolic blood pressures were measured from both arms and ankles (Microlife WatchBP Office Central, Microlife AG, Widnau, Switzerland),[20] providing information also on ankle brachial index.

Physical function measures included maximal and normal walking speed over 4 metres,[21] Short Physical Performance Battery (walking speed, balance and chair rise test)[22] and 1-leg standing balance.[23] In addition, maximal handgrip strength was measured on the dominant hand in a sitting position with Jamar dynamometer following protocol used in earlier large-scale ageing studies.[24]

Cognitive testing included Cambridge Neuropsychological Test Automated Battery (CANTAB), computerised test battery covering multiple cognitive domains including learning and memory, working memory, information processing and reaction time.[25] In addition, Trail Making Test,[26] Mini Mental State Examination and verbal recall from the Consortium to Establish a Registry for Alzheimer's Disease.[27]

During the clinical visit, the participants also self-completed depression (Beck's depression inventory)[28] and dietary questionnaires, which included questions about eating rhythm and indicators of healthy diet. In addition, participants were given a sexual health questionnaire to be filled at home, which was then returned back to the university. Details of these measurements are provided in online supplemental table 1.

Prior to the clinical examination, the participants received a wrist-worn triaxial ActiGraph wActiSleep-BT accelerometer via mail and were instructed to wear it continuously on their non-dominant wrist for 24 hours per day for at least 7 days and nights, including at least two workdays and two days off. Detailed description about the wrist-worn accelerometer data collection and data reduction procedures applied to the clinical substudy have been provided elsewhere.[19]

During the clinical examination visit, the participants were provided two additional devices to measure their movement behaviour and the study nurse instructed the use of these devices. While using the devices the participants were asked to fill in a daily diary, in which they provided information on their workdays and days off, and on their daily bedtimes and modes of commute. The study nurse fastened a triaxial Axivity AX3 accelerometer (Axivity, Newcastle, UK) to the participants' right thigh using adhesive waterproof film dressing. Participants were asked to wear the accelerometer for at least two workdays and two days off at all times when still working, and a minimum of any 4 days when retired. Fastening the accelerometer to thigh enables identification of postures such as sitting and standing and types of physical activity such as walking, running and cycling based on thigh angle and acceleration.[29 30] Detailed measurement protocol and accelerometer data reduction procedures for thigh-worn accelerometers have been described elsewhere.[31] In addition, participants received a waist-worn SenseDoc 2.0 device (Mobysens Technologies, Canada), which includes a Global Positioning System (GPS) sensor and a tri-axial accelerometer.[32] The combined GPS and accelerometer data provide information about intensity of the physical activity, but also about activity contexts and locations that is home, transportation, and allows linking environmental characteristics to these locations. Participants were requested to wear the device during waking hours for at least two workdays and two days off when still working, and a minimum of any 4 days when retired. Details of the measurements and data processing have been described earlier.[33 34] Participants were also requested to undergo a 24-hour ambulatory blood pressure measurement after the clinical visit.[20] The measurement was performed with a Microlife WatchBP O3 Monitor (Microlife AG, Widnau, Switzerland). Blood pressure was measured every 30 min over the entire 24 hours. Participants reported times of going to bed and waking up, which were used to calculate average awake and asleep blood pressure.

Within 1 month from the initial clinical examination visit, two additional examinations were conducted. Physical fitness tests (n=270) were performed by an experienced exercise physiologist at the Paavo Nurmi Centre, Sports & Exercise Medicine Unit in connection with University of Turku. Physical fitness testing included individually performed muscular fitness tests, modified push-up test and sit-up test, which are described in detail elsewhere.[19 35] In addition, cardiorespiratory fitness was measured in a group of 3–4 persons with indirect submaximal bicycle ergometer test according to the American College of Sports Medicine guidelines.[19 36]

In addition, arterial stiffness was assessed as carotid-femoral pulse wave velocity measurements (SphygmoCor PVx with MM3 electronic module and Millar tonometer), by study nurse at the university research facilities, which is described in detail elsewhere.[37] Details of the clinical measurements are provided in online supplemental table 1.

## Linkage to register data

Data collected from the study participants have been complemented with national register data and the linkage is conducted by using the national personal identification numbers (unique number assigned to all Finnish residents). Health-related outcomes are available until the end of 2019 from the hospital discharge information recorded by the Finnish Institute for Health and Welfare and the Cancer Register. In addition, medication purchase, special reimbursement, rehabilitation, sick-leave days are available from the registers of the Social Insurance Institution of Finland, and types and causes of workplace and commuting injuries from the Federation of Accident Insurance Institutions. Details of the register data are provided in online supplemental table 1.

Several indicators of the neighbourhood characteristics, such as a standardised index for socioeconomic disadvantage, degree of greenness and daily temperature in the neighbourhood, have been derived by using information on participants' geocoded residential addresses from the Population Register Centre of Finland and these data were positioned to the Statistics Finland Grid Database, a 250×250 m map grid that covers the whole country.[38] Details of these neighborhood-level indicators are provided in online supplemental table 1.

## Patient and public involvement

Study population or the general public were not involved in the planning, design or conduction of the study.

## FINDINGS TO DATE
### Movement behaviours

One of the focuses of the FIREA Study has been to understand how movement behaviours change when people transition from full time work to retirement. By using annually collected wrist-accelerometer data and by addressing differences between sex and occupations, we have shown that women retiring from manual occupations decrease their physically activity, whereas men from non-manual occupations slightly increase physical activity after retirement.[39] Despite the declining trend among women, they are still more active than men both before and after retirement. Our results also suggest that sleep duration and sedentary time, especially among women retiring from manual occupations, increase after retirement.[15 16 40 41] More recently, we have applied compositional data analysis, which allows examination of concomitant changes of the 24-hour movement behaviours, that is physical activity, sedentary time and sleep.[17] Our findings suggest that retirement decreases the proportion of time spent in active behaviours, especially moderate-to-vigorous physical activity, in relative to time spent in passive behaviours, such as sedentary time and sleep. These findings highlight that retirement is an important time period in late midlife for paying special attention to activity behaviour and develop new daily routines, which can replace work-related and commuting-related activities and involve physical activity in different forms. With advancing age, the importance of physically active lifestyle is emphasised, because it helps not only to prevent and treat non-communicable diseases, such as cardiovascular disease, type 2 diabetes and various cancers,[42] but it also helps to maintain physical functioning[43] and improve mental well-being.[44]

### Health and mental well-being

In addition to movement behaviours, we have also been interested in retirement-induced changes in various health and well-being indicators. We have shown that psychological distress decreases drastically after retirement and especially among those with poorer psychosocial working conditions and social living environment.[45] Similarly, life satisfaction increases, most prominently among women, those with suboptimal health and those living without a spouse.[46] Previous studies have reported both improvement and decline in self-reported health after retirement,[9 47–49] which was also confirmed by our findings. However, by using latent trajectory analysis to identify groups of individuals who show similar developmental trajectories over time, we found that a large majority of participants actually maintain their perceived health status during retirement transition and smaller subgroups of people show improvement or decline in perceived health.[50] Those whose health perception declined during the retirement transition had lower occupational status, physically strenuous work and job strain. Overall, our findings suggest that people with physically or psychosocially demanding work may improve their psychological well-being after retirement, but this is not necessarily reflected in their health perception. Our future work with the extended follow-up will reveal more about the long-term development of health, morbidity as well as physical and cognitive functioning in the post-retirement years.

### Predictors of extended employment

By using the information about timing of retirement, we have examined how factors related to health, work, as well as family and social relations predict retirement timing. We have observed that those who have better health and work ability,[51] good worktime control[52] and full-time working spouse[53] are more likely to extend their employment beyond the pensionable age compared with those who retire at their state pension age or earlier. We have also shown that men are more likely to extend their employment than women, and this is largely explained by men having the above mentioned characteristics more often.[54] We have also compared health development among those extending their employment to those who retire early or at their pensionable age and found that voluntary extension of the working career beyond pensionable age does not have either positive or negative effects on health and physical functioning among older workers.[55] Our results suggest that it would be important to develop and implement practices that support extended and flexible working life participation among older workers.

The complete list of publications can be found at the FIREA Study website (https://sites.utu.fi/firea/en/publications/).

## FUTURE PLANS

The second phase of the FIREA Study will take place in 2023–2025, when participants reach the age of 70 years. All participants who have participated to at least one survey, accelerometer measurement or clinical examination visit will be invited to the respective follow-up measurements. New information sheet and informed consent will be provided to the participants.

The questionnaire survey will include the same questions as previously, except questions related to work and retirement are removed and new questions relevant for the age group will be included, for example, weight history, chronotype, muscle strengthening and balance improving activities, functional ability, fear of falling, mental well-being, social relationship, loneliness, financial situation and working while retired. Activity substudy will be conducted using similar protocol and devices as in the previous study waves. The clinical substudy will also consist of the same measurements as in the previous study waves with few additions such as carotid artery ultrasound measurement and the 6-minute walk test, which will replace the submaximal bicycle ergometer test. The long-term aim is to conduct subsequent follow-ups every 5 years and follow participants into old age. In the

upcoming years and with the new data, the focus of the FIREA Study will shift more towards examining the determinants and development of health and functioning with an advancing age. It is of interest to study, for example, which factors associate maintenance of good physical and cognitive function after retirement.

## COLLABORATION

Further information about the FIREA Study can be found from the study website https://sites.utu.fi/firea/en/. The FIREA Study welcomes research collaboration proposals that fall within the general aims of the project. We follow the EU general data protection regulation (679/2016) and the Finnish Data Protection Act (1050/2018) in data management and use. Anonymised partial datasets of the FIREA Study are available by application with bona fide researchers with an established scientific record and bona fide organisations. For additional details, please contact principal investigator of the FIREA study (SS).

**Acknowledgements** The FIREA Study warmly thanks the study participants for their continuous commitment to the study. The research assistant Kati Pyykkö, study nurses Hanna Pynnönen and Minna Hyppönen, data manager Jesse Pasanen and other assisting personnel are acknowledged for their invaluable effort in the project.

**Contributors** SS and JV designed the cohort study. SS and KS drafted the manuscript. KS and JP performed the data analysis. SS, KS, TL, SM, AP, JP and JV critically revised the manuscript and approved the final manuscript. SS is guarantor.

**Funding** The FIREA Study has been financially supported by Academy of Finland (264944, 286294, 294154, 319246 and 332030); Ministry of Education and Culture; Hospital District of Southwest Finland, Finnish State Grants for Clinical Research; Finnish Work Environment Fund (118060); Juho Vainio Foundation; Signe and Ane Gyllenberg Foundation, Päivikki and Sakari Sohlberg Foundation and The Finnish Foundation for Cardiovascular Research.

**Competing interests** None declared.

**Patient and public involvement** Patients and/or the public were not involved in the design, or conduct, or reporting, or dissemination plans of this research.

**Patient consent for publication** Not applicable.

**Ethics approval** This study involves human participants and was approved. The FIREA Study has been approved by the Ethics Committee of Hospital District of Southwest Finland (84/1801/2014). Participants gave informed consent to participate in the study before taking part.

**Provenance and peer review** Not commissioned; externally peer reviewed.

**Data availability statement** Data are available upon reasonable request. Anonymised partial datasets of the FIREA study may be shared on a case-by-case basis upon request for bona fide researchers with an established scientific record and bona fide organisations. Data sharing outside the group is done in collaboration with FIREA group and requires a data-sharing agreement. Investigators can submit an expression of interest to FIREA research group at University of Turku, Finland ( firea@utu.fi).

**ORCID iDs**
Sari Stenholm http://orcid.org/0000-0001-7560-0930
Tuija Leskinen http://orcid.org/0000-0001-7499-6128
Anna Pulakka http://orcid.org/0000-0002-0602-8632
Jussi Vahtera http://orcid.org/0000-0002-6036-061X

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
