## [Reviewer comments · BMJ Open]

ARTICLE DETAILS

TITLE (PROVISIONAL)	Cohort Profile: Finnish Retirement and Aging study (FIREA), a prospective cohort study
AUTHORS	Stenholm, Sari; Suorsa, Kristin; Leskinen, Tuija; Myllyntausta, Saana; Pulakka, Anna; Pentti, Jaana; Vahtera, Jussi

VERSION 1 – REVIEW

REVIEWER	Ryan, Lindsay University of Michigan
REVIEW RETURNED	11-Aug-2023

GENERAL COMMENTS	The current paper outlines the design and purpose of the Finnish Retirement and Aging Study (FIREA), which aims to assess employed adults in Southwest Finland in the years before and after estimated statutory retirement dates. A focus of the current study is on assessing how retirement affects physical activity and health, ranging from cardiovascular health outcomes to mental and cognitive health. Below I outline clarification questions which if addressed will strengthen the current manuscript. 1. In the introduction, the authors note that at the time FIREA was initiated few published studies followed individuals throughout the retirement transition and that they are generally from non-generalizable study populations. There are two problems with this. First, there are actually many, many published papers reporting on individuals before, during and after retirement. There are too many to list in this review, but two that come to mind are Calvo, Sarkisian & Tamborini (2013) in JoG: Psych Sciences, and Rohwedder & Willis (2010) in the Journal of Economic Perspectives. I mention these two in particular to illustrate the second problem I have with this statement – these two papers are among many that use large, nationally representative panel study data. These come from the Health and Retirement Study family of studies, but there are others. The authors are correct that the published record shows contrasting findings and the addition of objective measures like actigraphy used in FIREA is an important contribution. This section needs to be revised to better reflect the publication record. 2. Related to the previous point, the authors make note that much of the published work comes from non-representative samples. Does this imply that the FIREA sample is representative? Were survey weights generated and used to make conclusions from this data relevant to a larger population? From my reading, the sample is large but not necessarily representative. Perhaps the authors could argue it generalizes to the employed adults in Southwest Finland who are nearing retirement? More clarification is needed. If claims about generalizability cannot be made, this needs to be clearly noted
---

	in the study limitations. 3. Regarding the study sample, can the authors please provide additional information about why Southwest Finland was the focus? For those readers not versed in the sociodemographics of Finland, this would be useful. It would also be helpful to include a very brief, one to two sentence explanation of the retirement/pension policy in Finland so readers can understand the implications of your study design.
--	--

REVIEWER	Neville, Charlotte Queen's University Belfast
REVIEW RETURNED	08-Sep-2023

GENERAL COMMENTS	The manuscript provides an overview of the Finnish Retirement and Ageing Study. This is an important article which will be referred to many times by the study research team in their future publications and also by other researchers who may be interested in using the data or applying some of the design or methodologies to their own research. The manuscript serves the important purpose of providing a detailed account of the design and execution of the study. While the paper is generally well written, there were some measures that could be more fully described. I have some suggestions for improvements in the detailed notes below. I am happy to recommend this paper for publication subject to minor revisions. List of revisions to address: Keywords: Consider including 'cohort study' or 'observational' as a keyword Abstract: The last sentence under the heading "Findings to date" is missing the word 'to' after 'certain behaviours such as sedentariness tend... to....increase.....' Strengths and limitations: Please also acknowledge the limitations associated with the self-reported nature of questionnaire data which is of course common to many ageing cohort studies. Apart from a healthy worker effect, please also acknowledge the under-representation of males (Table 1 shows that 82% of the cohort were female) and those who were single (71% were married or co-habiting) which would limit the generalisability of results. Introduction: Page 4: The authors state that 'although several studies on retirement and health had been published at the time FIREA was initiated, only few studies have been able to follow individuals through retirement transition.' This needs to be reworded and more references added to acknowledge the many ageing cohort studies that do follow individuals through the retirement transition e.g. the Health and Retirement Study (HRS) International Family of Studies https://hrs.isr.umich.edu/about/international-family-studies which have been conducted across the world. The HRS is the model for a growing network of longitudinal aging studies. Many of the studies in this network recruit participants from age 50 years onwards and follow them up through the transition to retirement and beyond using similar assessments i.e. clinical health assessments, questionnaires,
--

	activity sub-studies etc. Cohort description: Page 5: Study population and data collection: The aim was to gather data from at least two time points before and two time points after the transition to statutory retirement. The next sentence then states that 6783 responded to at least one questionnaire. How many participants achieved your aim and responded to at least two time points?? Page 5: Can more detail be provided on recruitment e.g. how were participants invited to take part in the initial study – was it by an advance letter or information sheet etc? How long did it take participants to complete each questionnaire? Please provide detail in the text. Did each follow up questionnaire include the same questions as baseline? Did participants receive any reimbursement for completing the questionnaire or taking part in the sub studies e.g. travel expenses to the University. Can you provide some indication as to the time frame (range) from participants completing the first questionnaire to taking part in the sub-studies. Page 6: Please provide information on how participants were invited to participate in the clinical sub-study – e.g. was it by a letter sent in the post? Page 6: The clinical sub-study was conducted at the University of Turku. It may be worth saying a little bit about why a University setting was used rather than a hospital based centre or home visit. Were all the clinical assessments conducted in the morning since participants had to provide a fasting blood sample? Page 6: ‘Up to five annual follow up clinical visits were conducted...’ – what was the response rate for this? – out of the 290 who consented, how many participants completed the follow-up visits? Page 6: The word ‘the’ needs to be added to the sentence beginning ‘However, this influenced only a small proportion of the participants, since... the...majority.....’ Page 7: The second phase of the study is taking place between 2023 and 2025. Can you provide any indication of what the second phase will include? – will it just be a follow up questionnaire survey using the same questionnaire or will new measures be introduced? Will a clinical assessment or activity assessment form part of the second phase? As part of the consent process in the first phase, were participants asked if they were happy to be revisited/ followed up for future waves/phases? Was separate consent obtained for the sub-studies? Clinical sub-study measurements Page 8: How long was the clinical assessment? Page 8: Participants responded to depression and dietary questionnaires – were these completed face to face or self-completed by the participant? Can you also provide some information on the dietary questionnaire – for example, was it a food frequency questionnaire, was it a validated questionnaire? Page 8: The authors state that a sexual health questionnaire was completed at home – please state how this questionnaire was returned - was it posted back? Supplementary Table 1 mentions a 24-hour ambulatory BP device – when was this carried out? Please provide more detail within the text.
--	--

	Page 8: Were participants offered feedback in relation to their blood glucose and lipid levels? If there were any incidental findings found during the health assessment or in the analysis of their bloods e.g. abnormal fasting glucose levels outside the expected normal range etc were these communicated back to the participant or was their General Practitioner informed? Page 8: In general, I think the section on 'clinical sub-study' measurements should include some more detail regarding the basic protocol for blood and hair sample handling and storage processing of samples (e.g. how soon after venepuncture were samples processed and analysed, where were samples stored/conditions etc). Although some of the methods are referenced within Table 1, it would be useful to include some more detail within the text, especially in relation to the blood samples, the hair sample, grip strength since they have not been detailed elsewhere. For example, what is meant by 'office' blood pressure? Is this the same as resting systolic and diastolic blood pressure? Hand grip strength – how many measurements were taken, were they taken from the dominant hand or non-dominant hand? Was an average of grip strength measurements used? Page 9: Physical fitness tests were performed within one month from the initial clinical examination visit. How many participants completed these tests? Where were these tests conducted? Patient and public involvement Page 10: Can the authors please comment as to why there was no PPI involvement? Will PPI be considered in future phases of the study? PPI is considered to be an important component of research studies and many funding bodies do request it. Although there is a sentence regarding the second phase of the study on page 7, it would be useful to have a separate section on 'Future works/Plans' and include some of the key areas of focus within ongoing research and how the data is being explored, as well as the overall plans for forthcoming publications or reports. Some of this is already stated in different parts of the text (for example page 12) but it may be better to have it under a separate heading. Page 13: In relation to Data availability, do the research team have an plans to deposit data in any International Data Repositories for Age related research? Table 1 – please define ATC in full. Supplementary Table 1: Under clinical measurements, what is meant by 'office blood pressure'? – is this the same as systolic and diastolic blood pressure which is the more commonly used terminology?
--	---

VERSION 1 – AUTHOR RESPONSE

Response to Reviewers' comments:

Reviewer 1

COMMENT 1: The current paper outlines the design and purpose of the Finnish

Retirement and Aging Study (FIREA), which aims to assess employed adults in

Southwest Finland in the years before and after estimated statutory retirement dates.

A focus of the current study is on assessing how retirement affects physical activity and health, ranging from cardiovascular health outcomes to mental and cognitive health. Below I outline clarification questions which if addressed will strengthen the current manuscript.

RESPONSE: We thank the Reviewer for the positive and useful comments.

Please see our point-by-point responses below.

COMMENT 2: In the introduction, the authors note that at the time FIREA was initiated few published studies followed individuals throughout the retirement transition and that they are generally from non-generalizable study populations. There are two problems with this. First, there are actually many, many published papers reporting on individuals before, during and after retirement. There are too many to list in this review, but two that come to mind are Calvo, Sarkisian & Tamborini (2013) in *JoG: Psych Sciences*, and Rohwedder & Willis (2010) in the *Journal of Economic Perspectives*. I mention these two in particular to illustrate the second problem I have with this statement – these two papers are among many that use large, nationally representative panel study data. These come from the Health and Retirement Study family of studies, but there are others. The authors are correct that the published record shows contrasting findings and the addition of objective measures like actigraphy used in FIREA is an important contribution. This section needs to be revised to better reflect the publication record.

RESPONSE: Thank you for this comment. We agree that large, existing cohort studies focusing on people aged 50 years and older needs to be better reflected in the Introduction. We have also added suggested references when mentioning earlier studies about retirement and health. We have now re-written parts of the Introduction section as follows:

“At the time FIREA study was initiated, several studies on retirement and health

had been published 1-4
, and there were several large cohort studies following
up participants from age 50 onwards, such as the Health and Retirement Study
(HRS) in the US 5
, the English Longitudinal Study on Aging (ELSA) in the UK 6
and the Survey of Health, Ageing and Retirement in Europe (SHARE) from
several European countries 7
. However, only a few studies had studied
changes in health behaviors by following individuals through retirement
transition. These studies had reported changes in health behaviors, such as
increased physical activity and improved sleep 8-12, but the results were
contrasting, and information on health and health behaviors was based on self-reported measures.”
(page 5)

COMMENT 3: Related to the previous point, the authors make note that much of the
published work comes from non-representative samples. Does this imply that the
FIREA sample is representative? Were survey weights generated and used to make
conclusions from this data relevant to a larger population? From my reading, the
sample is large but not necessarily representative. Perhaps the authors could argue
it generalizes to the employed adults in Southwest Finland who are nearing
retirement? More clarification is needed. If claims about generalizability cannot be
made, this needs to be clearly noted in the study limitations.

RESPONSE: We have removed the comment that many previous studies are
based on non-representative samples, because that is not clearly the case for
example with the HRS and ELSA. Although our study population represents
well the public sector workers who continue working beyond the age of 60
years, it is not generalizable to the whole population in that age group. We have
now better addressed factors that limit generalizability of the findings in the
“Strength and limitations” section as follows:

“The main limitation of the FIREA study is a healthy worker effect as the inclusion criteria include participation in the working life after the age of 60.

Those retired due to illness, disability or unemployed are not included in the study population. Moreover, men and individuals living alone are underrepresented, which further limits the generalizability of the results. (page 4)

COMMENT 4: Regarding the study sample, can the authors please provide additional information about why Southwest Finland was the focus? For those readers not versed in the sociodemographics of Finland, this would be useful. It would also be helpful to include a very brief, one to two sentence explanation of the retirement/pension policy in Finland so readers can understand the implications of your study design.

RESPONSE: We are happy to clarify. In fact, the eligible study population of the FIREA cohort comprised of public sector employees who were working in any of the a priori selected 10 towns or five hospital districts around Finland OR in any town in Southwest Finland. We included these 10 towns and all hospital districts, because they have been involved in a large Finnish Public Sector (FPS) study since 1997. The FIREA clinical substudy was conducted only among employees working in Southwest Finland due to practical reasons, because these employees lived within 100 kilometers from the University of Turku, where the clinical measurement took place.

As suggested by the Reviewer, we have also included a short description of the retirement system in Finland as follows:

“In Finland, from 2005 onwards, public sector employees have been able to retire on a statutory basis after the age of 63 years, but at the latest before the age of 68 years. Following a pension reform in 2017, each age group has their own retirement age, which is tied to the life expectancy. Postponing retirement from the estimated retirement age will accrue pension income level. After the pension reform, employees may also retire on partial early old-age pension,

years of-service pension (for those who have worked for at least 38 years in job that requires great mental or physical effort), or on full or partial disability pension 13. The FIREA study participants were, due to the inclusion criteria, mostly retiring due to old age with a mean age of 64.0 years (SD 1.4).” (pages 8-9)

Reviewer 2

COMMENT 1: The manuscript provides an overview of the Finnish Retirement and Ageing Study. This is an important article which will be referred to many times by the study research team in their future publications and also by other researchers who may be interested in using the data or applying some of the design or methodologies to their own research.

The manuscript serves the important purpose of providing a detailed account of the design and execution of the study. While the paper is generally well written, there were some measures that could be more fully described. I have some suggestions for improvements in the detailed notes below.

I am happy to recommend this paper for publication subject to minor revisions.

RESPONSE: We thank the Reviewer for the positive and constructive comments, which helped us to improve the manuscript.

COMMENT 2: Keywords: Consider including ‘cohort study’ or ‘observational’ as a keyword.

RESPONSE: We have added ‘cohort study’ in the key word list as suggested.

COMMENT 3: Abstract: The last sentence under the heading “Findings to date” is missing the word ‘to’ after ‘certain behaviours such as sedentariness tend... to...increase.....’

RESPONSE: We have corrected the sentence as suggested.

COMMENT 4: Strengths and limitations: Please also acknowledge the limitations associated with the self-reported nature of questionnaire data which is of course

common to many ageing cohort studies.

RESPONSE: We have added potential response bias related the survey instruments as a limitation to the ‘Strengths and limitations’ section as follows:

“In addition to survey measures, which are prone to response bias, participants have been followed with accelerometer and clinical measurements, which are more accurate and enable capturing subtle changes in movement behaviors and cardiometabolic risk markers.” (page 4)

COMMENT 5: Apart from a healthy worker effect, please also acknowledge the under-representation of males (Table 1 shows that 82% of the cohort were female) and those who were single (71% were married or co-habiting) which would limit the generalisability of results.

RESPONSE: We have commented under-representation of men and persons living alone in the ‘Strengths and limitations’ section as follows:

“Moreover, men and persons living alone are under-represented, which further limits the generalizability of the results.” (page 4)

COMMENT 6: Page 4: The authors state that ‘although several studies on retirement and health had been published at the time FIREA was initiated, only few studies have been able to follow individuals through retirement transition.’ This needs to be reworded and more references added to acknowledge the many ageing cohort studies that do follow individuals through the retirement transition e.g. the Health and Retirement Study (HRS) International Family of Studies <https://hrs.isr.umich.edu/about/international-family-studies> which have been conducted across the world. The HRS is the model for a growing network of longitudinal aging studies. Many of the studies in this network recruit participants from age 50 years onwards and follow them up through the transition to retirement and beyond using similar assessments i.e. clinical health assessments, questionnaires, activity sub-studies etc.

RESPONSE: Thank you for this comment. We agree that large, existing cohort studies focusing on people aged 50 years and older needs to be better reflected in the Introduction. We have now re-written parts of the Introduction section as follows:

“At the time FIREA study was initiated, several studies on retirement and health had been published 1-4

, and there were several large cohort studies following

up participants from age 50 onwards, such as the Health and Retirement Study (HRS) in the US 5

, the English Longitudinal Study on Aging (ELSA) in the UK 6

and the Survey of Health, Ageing and Retirement in Europe (SHARE) from several European countries 7

. However, only few studies had studied changes

in health behaviors by following individuals through retirement transition. These studies had reported changes in health behaviors, such as increased physical activity and improved sleep 8-12, but the results were contrasting, and information on health and health behaviors was based on self-reported measures.” (page 5)

COMMENT 7: Page 5: Study population and data collection: The aim was to gather data from at least two time points before and two time points after the transition to statutory retirement. The next sentence then states that 6783 responded to at least one questionnaire. How many participants achieved your aim and responded to at least two time points??

RESPONSE: 5506 participants responded at least in two time points and of them 3948 provided response before and after retirement as aimed for. This is now clarified in the manuscript as follows:

“In total, 6,783 participants had responded to at least one questionnaire by the

end of 2018. 5,506 participants responded at least in two time points and of them 3,948 provided response before and after retirement.” (page 6)

COMMENT 8: Page 5: Can more detail be provided on recruitment e.g. how were participants invited to take part in the initial study – was it by an advance letter or information sheet etc?

RESPONSE: The individuals were first identified from the register maintained by pension insurance institute for the municipal sector in Finland. Individuals whose estimated retirement date was between 2014 and 2019 were invited to participate in the FIREA study by sending them a questionnaire 18 months prior to their estimated retirement date. The questionnaire included information sheet and informed consent. This is now clarified as follows:

“Individuals whose estimated retirement date was between 2014 and 2019, were invited to participate in the FIREA study by sending them a questionnaire 18 months prior to their estimated retirement date (N=10,629). The questionnaire included information sheet and informed consent.” (page 6)

COMMENT 9: How long did it take participants to complete each questionnaire? Please provide detail in the text.

RESPONSE: The printed version of survey questionnaire included 20 pages and it took about 20-30 minutes to complete.

This information is now provided in the manuscript:

“The questionnaire was 20 pages long and it took about 20-30 minutes to complete.” (page 9)

COMMENT 10: Did each follow up questionnaire include the same questions as baseline?

RESPONSE: Yes, the content of the questionnaire remained similar in the first phase of the study (2013-2022).

This is now clarified as follows:

“The content of the questionnaire was similar in each follow-up year.” (page 9)

COMMENT 11: Did participants receive any reimbursement for completing the questionnaire or taking part in the sub studies e.g. travel expenses to the University.

RESPONSE: Participants did not receive any reimbursement for completing the questionnaire or taking part in the sub studies. This information is now provided in the manuscript:

“Participants did not receive any reimbursement for taking part in the study.”

(page 7)

COMMENT 12: Can you provide some indication as to the time frame (range) from participants completing the first questionnaire to taking part in the sub-studies.

RESPONSE: The first wrist accelerometry was conducted on average 2.1 months (SD 1.2) after responding to the questionnaire. The first clinical visit was conducted on average 4.3 months (SD 3.4) after responding to the questionnaire. The explanation for the long interval between these measurements is that after receiving the questionnaires, we invited the respondents to participate either to activity sub-study or clinical sub-study by mailing them a separate information sheet and informed consent. Those who returned the informed consent via mail were then sent an accelerometer via mail or their clinical visit date was scheduled.

This is now clarified as follows:

“Activity substudy: “908 consented to participate and they were sent the accelerometer on average 2.1 months (SD 1.2) after completing the questionnaire.” (page 7)

Clinical substudy: “The first visit took place on average 4.3 months (SD 3.4) after completing the questionnaire.” (page 7)

COMMENT 13: Page 6: Please provide information on how participants were invited to participate in the clinical sub-study – e.g. was it by a letter sent in the post?

RESPONSE: After responding to the survey questionnaire, participants were mailed a separate information sheet and informed consent. This is now explained in the manuscript as follows:

“Finnish-speaking survey cohort participants who were still working, had an estimated statutory retirement date between 2017 and 2019 and who lived in the Southwest Finland were invited to participate by mailing them a separate information sheet and informed consent (n=773).” (page 7)

COMMENT 14: Page 6: The clinical sub-study was conducted at the University of Turku. It may be worth saying a little bit about why a University setting was used rather than a hospital based centre or home visit.

RESPONSE: In the Faculty of Medicine at the University of Turku, there is a special research facility to conduct population-based health examination studies. In the beginning of the study participants were still in the working life and well-functioning, thus home visits were not needed. Moreover, home visits would have not been feasible, since participants lived in a large geographic area in the Southwest Finland. The research facilities are now explained in the manuscript as follows:

“Annual clinical examination visits were conducted at the University of Turku, in the Faculty of Medicine’s research facilities.” (page 10)

COMMENT 15: Were all the clinical assessments conducted in the morning since participants had to provide a fasting blood sample?

RESPONSE: Yes, clinical assessments were conducted in the morning. This is now specified in the manuscript as follows:

“Measurements were conducted in the morning by trained study nurses” (page 10).

COMMENT 16: Page 6: ‘Up to five annual follow up clinical visits were conducted...’ – what was the response rate for this? – out of the 290 who consented, how many

participants completed the follow-up visits?

RESPONSE: Out of the 290 participants, 274 completed three annual

measurements (94%). Since the aim was to measure participants also postretirement, only those who had not retired by the third follow-up visit, were

invited to the fourth visit (35 participants) and to the fifth visit (3 participants).

We did not have resources to conduct several post-retirement measurements

to everyone. This is now clarified in the manuscript as follows:

“The mean number of annual measurements in the clinical sub-study was 3.0

(range 1–5, SD 0.6).” (page 8)

COMMENT 17: Page 6: The word ‘the’ needs to be added to the sentence beginning

‘However, this influenced only a small proportion of the participants, since...

the...majority.....’

RESPONSE: We have corrected the sentence as suggested.

COMMENT 18: Page 7: The second phase of the study is taking place between

2023 and 2025. Can you provide any indication of what the second phase will

include? – will it just be a follow up questionnaire survey using the same

questionnaire or will new measures be introduced? Will a clinical assessment or

activity assessment form part of the second phase?

RESPONSE: We are happy to specify the content of the new study wave. We

have now described in detail the upcoming changes in the data collection as

follows:

“The questionnaire survey will include the same questions as previously,

except questions related to work and retirement are removed and new

questions relevant for the age group will be included, for example weight

history, chronotype, muscle strengthening and balance improving activities,

functional ability, fear of falling, mental wellbeing, social relationship,

loneliness, financial situation and working while retired. Activity sub-study will

be conducted using similar protocol and devices as in the previous study

waves. The clinical sub-study will also consist of the same measurements as in the previous study waves with few additions such as carotid artery ultrasound measurement, Montreal Cognitive Assessment (MoCA) test and the 6-minute walk test, which will replace the submaximal bicycle ergometer test.” (pages 16-17)

COMMENT 19: As part of the consent process in the first phase, were participants asked if they were happy to be revisited/followed up for future waves/phases? Was separate consent obtained for the sub-studies?

RESPONSE: Separate consent was obtained for the activity and clinical substudy in the first phase of the FIREA study. Participants were also informed that the study will continue later and they will then be invited to participate. They are provided a new information sheet and informed consent in the second study phase. This is now clarified in the manuscripts as follows: “New information sheet and informed consent will be provided to the participants.” (page 16)

COMMENT 20: Page 8: How long was the clinical assessment?

RESPONSE: The clinical examination visit took about 2.5-3 hours depending on the participant. This is now specified in the manuscript: “Measurements were conducted in the morning by trained study nurses and the visit lasted about 2.5–3 hours.” (page 10)

COMMENT 21: Page 8: Participants responded to depression and dietary questionnaires – were these completed face to face or self-completed by the participant? Can you also provide some information on the dietary questionnaire – for example, was it a food frequency questionnaire, was it a validated questionnaire?

RESPONSE: The questionnaires were self-completed during the study visit. We did not use food frequency questionnaire, but inquired eating rhythm and indicators of healthy diet (e.g. vegetables, fruit, fat free products). This is now

specified in the manuscript as follows:

“During the clinical visit, the participants also self-completed depression (Beck’s depression inventory) 12 and dietary questionnaires, which included questions about eating rhythm and indicators of healthy diet.” (page 11)

COMMENT 22: Page 8: The authors state that a sexual health questionnaire was completed at home – please state how this questionnaire was returned - was it posted back?

RESPONSE: Yes, it was returned back to university in a sealed envelope.

This is now specified in the manuscript as follows:

“In addition, participants were given a sexual health questionnaire to be filled at home, which was then returned to the university.” (page 11)

COMMENT 23: Supplementary Table 1 mentions a 24-hour ambulatory BP device – when was this carried out? Please provide more detail within the text.

RESPONSE: The device was given at the clinical visit and the measurement was performed during the next 24 hours. This is now explained in the manuscript as follows:

“Participants were also requested to undergo a 24-hour ambulatory blood pressure measurement after the clinical visit. 20 The measurement was performed with a Microlife WatchBP O3 Monitor (Microlife AG, Widnau, Switzerland). Blood pressure was measured every 30 minutes over the entire 24 hours. Participants reported times of going to bed and waking up, which were used to calculate average awake and asleep blood pressure.” (page 12)

COMMENT 24: Page 8: Were participants offered feedback in relation to their blood glucose and lipid levels? If there were any incidental findings found during the health assessment or in the analysis of their bloods e.g. abnormal fasting glucose levels outside the expected normal range etc were these communicated back to the participant or was their General Practitioner informed?

RESPONSE: The blood samples were aliquoted at the same day and then stored at -80°C. To maintain the comparability across the follow-up years, all analyses were performed simultaneously at the laboratory of Turku University Hospital after the data collection for the first phase of the study collection was completed in 2022. Therefore, participants receive retrospectively the results of their blood samples.

From all other measurements, in which the results were immediately available, we provided feedback to the participants during the clinical visits. In case of e.g. abnormal blood pressure values, the participants were instructed to contact a general practitioner. We have added participant feedback to the manuscript as follows:

“Participants received feedback of the measurements for which results were available.” (page 10)

COMMENT 25: Page 8: In general, I think the section on ‘clinical sub-study’ measurements should include some more detail regarding the basic protocol for blood and hair sample handling and storage processing of samples (e.g. how soon after venepuncture were samples processed and analysed, where were samples stored/conditions etc). Although some of the methods are referenced within Table 1, it would be useful to include some more detail within the text, especially in relation to the blood samples, the hair sample, grip strength since they have not been detailed elsewhere. For example, what is meant by ‘office’ blood pressure? Is this the same as resting systolic and diastolic blood pressure? Hand grip strength – how many measurements were taken, were they taken from the dominant hand or nondominant hand? Was an average of grip strength measurements used?

RESPONSE: Due to the large number of measurements, we initially compiled information of the methods and references only to the Supplement Table 1.

As suggested, we have now expanded description of clinical visit and its measurements as follows:

“A 10-hour fasting blood sample were drawn by venipuncture in the morning shortly after arriving to the clinic. All blood samples were aliquoted at the same day and then stored at -80°C. Plasma cholesterol (total, low-density lipoprotein cholesterol, high-density lipoprotein cholesterol), triglycerides, fasting insulin and glucose and C-reactive protein were analyzed later at the laboratory of Turku University Hospital, Finland.

For the hair sample a strand of hair was cut from a standardized area of the posterior vertex region of the head as close to the scalp as possible 12 Hair samples were stored in foil in a dry place protected from light. Hair cortisol and cortisone were analyzed later using mass spectrometry (Technical University of Dresden, Germany).

For anthropometrics, weight, height, body mass index and waist circumference were measured by using standard methods. Body composition was measured with Inbody 720 (Biospace Co., Seoul, Korea) 13. Systolic and diastolic blood pressure was measured from both arms and ankles (Microlife WatchBP Office Central, Microlife AG, Widnau, Switzerland) 14, providing information also on ankle brachial index.

Physical function measures included maximal and normal walking speed over 4 meters 15, Short Physical Performance Battery (walking speed, balance and chair rise test) 16 and 1-leg standing balance 17. In addition, maximal handgrip strength was measured on the dominant hand in a sitting position with Jamar dynamometer following protocol used in earlier large-scale aging studies 18

Cognitive testing included Cambridge Neuropsychological Test Automated Battery (CANTAB®), computerized test battery covering multiple cognitive domains including learning and memory, working memory, information processing, and reaction time 19. In addition, Trail Making Test 20, Mini Mental

State Examination and verbal recall from the Consortium to Establish a Registry for Alzheimer's Disease (CERAD) 21

.” (page 10-11)

Regarding the question about “office blood pressure”, we used the term to indicate that the blood pressure measurement was conducted during clinical visit and not on participants' home. We have removed that term to avoid confusion.

Regarding the question about hand grip strength measurement. It was conducted on a dominant hand. The maximal test was repeated twice and if the results differed more than 10%, a third attempt was conducted. The best result is chosen for the analysis. The measurement protocol is used in several earlier large-scale aging studies.

COMMENT 26: Page 9: Physical fitness tests were performed within one month from the initial clinical examination visit. How many participants completed these tests? Where were these tests conducted?

RESPONSE: In total 270 participants participated in the fitness tests. Others had contraindications or were unwilling to do the tests. The tests were conducted at the Paavo Nurmi Centre, Sports & Exercise Medicine Unit in connection with University of Turku. We have amended the description of the physical fitness tests as follows:

“Within one month from the initial clinical examination visit, physical fitness tests (n=270) were performed by an experienced exercise physiologist at the Paavo Nurmi Centre, Sports & Exercise Medicine Unit in connection with University of Turku Physical fitness testing included individually performed muscular fitness tests, modified push-up test and sit-up test, which are described in detail elsewhere 13, 29. In addition, cardiorespiratory fitness was measured in a group of 3–4 persons with indirect submaximal bicycle

ergometer test according to the American College of Sports Medicine guidelines 13,30
.” (pages 12-13)

COMMENT 27: Page 10: Can the authors please comment as to why there was no PPI involvement? Will PPI be considered in future phases of the study? PPI is considered to be an important component of research studies and many funding bodies do request it.

RESPONSE: The FIREA study was established in 2013 and at that time PPI involvement in planning research projects was not as common practice in Finland as it is currently. We will consider PPI in the future phases of the FIREA study.

COMMENT 28: Although there is a sentence regarding the second phase of the study on page 7, it would be useful to have a separate section on ‘Future works/Plans’ and include some of the key areas of focus within ongoing research and how the data is being explored, as well as the overall plans for forthcoming publications or reports. Some of this is already stated in different parts of the text (for example page 12) but it may be better to have it under a separate heading.

RESPONSE: Thank you for this suggestion. We have now moved the description of the second phase of the study under heading “Future plans” and added future research topics.

“In the upcoming years and with the new data the focus of the FIREA study will shift more towards examining the determinants and development of health and functioning with an advancing age. It is of interest to study for example which factors associate maintenance of good physical and cognitive function after retirement.” (page 17)

COMMENT 29: Page 13: In relation to Data availability, do the research team have an plans to deposit data in any International Data Repositories for Age related

research?

RESPONSE: The FIREA dataset comprises health related participant data and their use is therefore restricted under the regulations on professional secrecy (Act on the Openness of Government Activities, 612/1999) and on sensitive personal data (Personal Data Act, 523/1999, implementing the EU data protection directive 95/46/EC). Due to these legal restrictions, the data from the FIREA study cannot be stored in public repositories or otherwise made publicly available. However, data access may be permitted on a case by case basis upon request for bona fide researchers with an established scientific record and bona fide organisations.

COMMENT 30: Table 1 – please define ATC in full.

RESPONSE: ATC stands for Anatomical Therapeutic Chemical Classification and is now defined in Supplementary Table 1.

COMMENT 31: Supplementary Table 1: Under clinical measurements, what is meant by ‘office blood pressure’? – is this the same as systolic and diastolic blood pressure which is the more commonly used terminology?

RESPONSE: As mentioned earlier, we used the term “office blood pressure” to indicate that the blood pressure measurement was conducted during clinical visit and not on participants’ home. We have removed that term to avoid confusion.

VERSION 2 – REVIEW

REVIEWER	Neville, Charlotte Queen's University Belfast
REVIEW RETURNED	31-Oct-2023
GENERAL COMMENTS	The authors have adequately addressed all my queries. Thank you.